# An Emulation of Randomized Trials of Administrating Antipsychotics in PTSD Patients for Outcomes of Suicide-Related Events

**DOI:** 10.3390/jpm11030178

**Published:** 2021-03-04

**Authors:** Noah R. Delapaz, William K. Hor, Michael Gilbert, Andrew D. La, Feiran Liang, Peihao Fan, Xiguang Qi, Xiaojiang Guo, Jian Ying, Dara Sakolsky, Levent Kirisci, Jonathan C. Silverstein, Lirong Wang

**Affiliations:** 1University of Pittsburgh School of Pharmacy, Pittsburgh, PA 15213, USA; nrd33@pitt.edu (N.R.D.); wkh7@pitt.edu (W.K.H.); mig64@pitt.edu (M.G.); adl64@pitt.edu (A.D.L.); fel41@pitt.edu (F.L.); 2Department of Pharmaceutical Sciences, Computational Chemical Genomics Screening Center, University of Pittsburgh School of Pharmacy, Pittsburgh, PA 15213, USA; pef14@pitt.edu (P.F.); xiq24@pitt.edu (X.Q.); xig53@pitt.edu (X.G.); 3Department of Internal Medicine, University of Utah, Salt Lake City, UT 84132, USA; jian.ying@hsc.utah.edu; 4Department of Psychiatry, University of Pittsburgh School of Medicine, Pittsburgh, PA 15213, USA; sakolskydj@upmc.edu; 5Department of Pharmaceutical Sciences, University of Pittsburgh School of Pharmacy, Pittsburgh, PA 15213, USA; 6Department of Biomedical Informatics, University of Pittsburgh School of Medicine, Pittsburgh, PA 15206, USA

**Keywords:** antipsychotics, PTSD, suicide, suicide-related behaviors, clinical trial emulation

## Abstract

Post-traumatic stress disorder (PTSD) is a prevalent mental disorder marked by psychological and behavioral changes. Currently, there is no consensus of preferred antipsychotics to be used for the treatment of PTSD. We aim to discover whether certain antipsychotics have decreased suicide risk in the PTSD population, as these patients may be at higher risk. A total of 38,807 patients were identified with a diagnosis of PTSD through the ICD9 or ICD10 codes from January 2004 to October 2019. An emulation of randomized clinical trials was conducted to compare the outcomes of suicide-related events (SREs) among PTSD patients who ever used one of eight individual antipsychotics after the diagnosis of PTSD. Exclusion criteria included patients with a history of SREs and a previous history of antipsychotic use within one year before enrollment. Eligible individuals were assigned to a treatment group according to the antipsychotic initiated and followed until stopping current treatment, switching to another same class of drugs, death, or loss to follow up. The primary outcome was to identify the frequency of SREs associated with each antipsychotic. SREs were defined as ideation, attempts, and death by suicide. Pooled logistic regression methods with the Firth option were conducted to compare two drugs for their outcomes using SAS version 9.4 (SAS Institute, Cary, NC, USA). The results were adjusted for baseline characteristics and post-baseline, time-varying confounders. A total of 5294 patients were eligible for enrollment with an average follow up of 7.86 months. A total of 157 SREs were recorded throughout this study. Lurasidone showed a statistically significant decrease in SREs when compared head to head to almost all the other antipsychotics: aripiprazole, haloperidol, olanzapine, quetiapine, risperidone, and ziprasidone (*p* < 0.0001 and false discovery rate-adjusted *p* value < 0.0004). In addition, olanzapine was associated with higher SREs than quetiapine and risperidone, and ziprasidone was associated with higher SREs than risperidone. The results of this study suggest that certain antipsychotics may put individuals within the PTSD population at an increased risk of SREs, and that careful consideration may need to be taken when prescribed.

## 1. Introduction

Post-traumatic stress disorder (PTSD) is a psychiatric disorder that can develop after an individual is exposed to a traumatic event, characterized by hypervigilance, disturbed sleep, avoidance of triggers, and alteration of mood, cognition, and behavior and is associated with high disability in older Americans [1]. The prevalence of PTSD in adults is high, as more than 70% adults worldwide will experience a traumatic event in their lives [2]. 

There are different treatment options for PTSD [3]. Aside from pharmacologic intervention, the mainstay for therapy includes cognitive behavioral therapy and other psychological treatments [3]. The road to recovery for PTSD patients is challenging, and most often requires a combination of treatment strategies [3]. Current guidelines recommend antidepressants, particularly paroxetine and sertraline, as first-line pharmacological treatment [3,4]. In addition, prazosin, an α1-blocker has been reported to be significantly more efficacious than placebo in reducing distressing dreams in PTSD patients [5] and sleep disturbances might be associated with suicidality in PTSD [6]. However, a large multisite randomized controlled trial of military veterans with chronic PTSD showed prazosin to be ineffective at reducing distressing dreams or improve sleep quality [7]. 

PTSD has been associated with suicide-related events (SREs) [8], though the extent and validity of the relationship are still to be confirmed [2]. In a study among Iraq and Afghanistan U.S. veterans, current depressive symptoms, PTSD, and history of prior traumatic brain injury (TBI) were all significantly associated with current suicidal ideation [9]. Pre-deployment insomnia is also associated with post-deployment PTSD and suicidal ideation in US Army soldiers [10]. Defined by suicidal ideation, attempt, and/or death from suicide, SRE risk is lower in PTSD patients than that of depression [2]. Though PTSD is shown to increase risk of suicidal ideation and attempt, a meta-analysis of 50 articles did not see an increase in death by suicide [11]. 

A prospective longitudinal study published in 2013 aimed to identity risk factors for suicide in current and former US military personnel [12]. LeardMann and colleagues discovered that risk factors for suicide include males, depression, manic–depressive disorder, and alcohol dependence, confirmed by several Cox proportional hazard models [12].

In our recent publication on the study on SREs in PTSD patients with bipolar disorder, we found that use of antipsychotics is a good feature for SREs prediction [13]. Antipsychotics are a class of medications indicated for the treatment of psychotic symptoms such as hallucinations, delirium, and mania [14]. Schizophrenia, depression, and bipolar disorders are conditions commonly thought to contribute to these symptoms [14]. First- and second-generation antipsychotics make up the two types of medications in the class. First-generation antipsychotics (FGAs), also known as typical antipsychotics, are observed to have a higher risk of extrapyramidal symptoms (EPS), such as acute dystonia, Parkinsonism, and tardive dyskinesia [15].

Typical antipsychotics include chlorpromazine, perphenazine, and haloperidol. This class of drugs mainly block dopamine D2 receptors in the brain, which provides the symptomatic treatment of the psychotic disorders [16]. The relatively low selectivity and nonspecific localization of these drugs cause the EPS, specifically through blockage of the mesolimbic pathway, that are common with the first-generation [15]. 

Atypical antipsychotics, known as second-generation antipsychotics (SGAs), are also used routinely in the treatment for PTSD [17]. Similarly to FGAs, SGAs block D2 receptors along with serotonin 5-HT2A receptors [18]. SGAs are associated with less EPS compared to FGAs. This is believed to occur due to SGAs transiently binding to D2 receptors along with rapid dissociation allowing for normal dopamine neurotransmission along with their higher affinity to serotonin receptors [18]. However, the use of SGAs is not without caution. Metabolic syndrome is a commonly reported side effect that includes a collection of syndromes such as stroke, diabetes, and cardiovascular disease [19].

### 1.1. Relationship between Antipsychotics and PTSD 

Antipsychotics, in the context of treatment for PTSD, have typically been used to augment the effects of first-line agents such as serotonin receptor inhibitors (SRIs) or as second-line treatment [20]. Small randomized controlled trials have been conducted to compare the efficacy of monotherapy SGAs, such as quetiapine and olanzapine, in patients with PTSD [21,22]. Results were limited but show promising efficacy for the treatment of PTSD with SGAs. 

Currently, there is no consensus of preferred antipsychotics to be used for the treatment of PTSD. The goal of this observational study is to identify specific antipsychotics that may be associated with a decreased incidence of SREs as defined above. 

### 1.2. Emulating Randomized Controlled Trial 

Creating guidance for clinical guidelines requires sound evidence on the effects of their recommendations on relevant outcomes. The golden standard of investigation relies heavily on randomized controlled trials. However, randomized controlled trials are not always feasible due to money, time, or ethical constraints. Observational data are often used as an alternative to randomized controlled trials. 

Studies using observational data are subject to selection and immortal-time bias, which can be controlled in randomized controlled trials [23]. In this study, we use emulation methods to control these two biases. Emulation attempts to eliminate selection bias by specifying a target trial and creating an appropriate protocol [24]. This allows for the comparison of individuals newly initiated on the medication against those who were never initiated. Immortal-time bias can be minimized by analyzing the data so that time of eligibility for treatment and time when treatment is initiated is the same [25].

## 2. Methods

### 2.1. Data Source

We accessed data from January 2004 to October 2019 from the Neptune system at University of Pittsburgh which manages use of patient electronic medical records from the UPMC health system for research purposes (rio.pitt.edu/services accessed on 3 March 2021). The database includes demographic information, diagnoses, encounters, medication prescriptions, prescription fill history, and laboratory tests. 

### 2.2. Inclusion/Exclusion Criteria and Endpoints/Follow Up 

The baseline eligibility criteria include initiated an antipsychotic after the diagnosis of PTSD (qualifying event), no history of suicidal-related events prior to the enrollment, no prior use of antipsychotics within one year, and at least one year of history in the electronic medical record. If eligibility criteria were met, patients were followed to the onset of the first suicide-related event (primary outcome) or until loss to follow up (stopping use of antipsychotic, switching to another antipsychotic, patients data being no longer accessible or reaching the end time of this study). 

### 2.3. Target Trials

In order to be included in the trial, eligible PTSD patients must have no experience of SREs and no usage of any antipsychotic during one year prior to the enrollment. Patients were randomly assigned to one of the compared antipsychotic arms in the target trial. Antipsychotics of interest include aripiprazole, haloperidol, lurasidone, olanzapine, quetiapine, risperidone, perphenazine, and ziprasidone. Patients could take any other medications for the treatment of their comorbidities. The trial will be stopped if a patient stops using the drug assigned or switches or starts using another antipsychotic, loses follow up, or experiences SREs. The outcome evaluated is the onset of SREs.

### 2.4. Emulating Target Trials

We attempted to emulate randomized controlled trials similarly to the work of Danaei and colleagues [26]. Confounding variables had to be measured at least once during the study duration. These variables can be accessed below in Table 1 of methods section and categories are based on ICD9 and ICD10 codes described in Appendix A. In addition to the baseline eligibility criteria that is discussed above, participants needed one or more years of continuous recording in the UPMC medical records and at least one medical visit within one year of initiation of trial. 

Monthly trials were collected from the UPMC EMR database from January 2004 to October 2019 (190 months of interval). Patients could be included in this study multiple times if a one-year washout period of antipsychotic was fulfilled and all inclusion criteria were met. Those eligible were placed in a specific target trial arm based on the antipsychotic used. The target trials of antipsychotics were then compared with the primary outcome being SREs experienced. Study duration was stopped if medication of interest was discontinued or if the patient’s EMR data were no longer available (loss to follow up or death). In other words, patients were right censored if they had no SREs at the end of the study period (i.e., administrative censoring), if a person failed to return for a study visit (i.e., lost to follow up), if they stopped using the antipsychotic of interest (no records of use within 3 months), or if they switched to another antipsychotic.

### 2.5. Per-Protocol Analysis 

Per-protocol analysis was utilized which requires that all the patients fully completed the given protocol regime. The effect of the two drugs was compared on the outcomes of two cohorts who completed the treatment originally allocated. This analysis may give rise to bias by baseline confounders and post-baseline, time-varying confounders. An approach suggested by Danaei and colleagues used a pooled logistic regression model in order to estimate the effect of treatment [25]. They used inverse probability weighting to create a population where treatment is independent of prognostic factors history. The baseline information included 12 categories of mental disorders (Appendix B) [27], age, gender, number of emergency department visits within one year prior to the enrollment and the summary can be found in the Table 1. To test the effects of concomitant medications, we performed an additional analysis by adjusting the most frequently used drugs of the central nervous system in drug pair(s) of interest.

In our emulation, a pooled logistic regression model was fitted to adjust the censoring effects. Switching from one of the paired antipsychotics to the other one was not considered since we believe this situation can be represented by the censor model. 

In addition, we used robust variances to calculate conservative 95% confidence intervals and truncated the inverse-probability weights to their 99th percentile. Those are options provided by the code from www.hsph.harvard.edu/causal/software (accessed on 27 November 2020). We also applied the Firth option in the logistic regression to accommodate the presence of rare events or complete separation. Datasets were prepared by Python [28] and the final analyses were carried out using SAS 9.4 (SAS Institute Inc., Cary, NC, USA) [29]. False discovery rate (FDR) q value is controlled to 0.05 to avoid the inflation of type I error cause by multiple hypothesis test. The calculation of FDR is conduct by R [30] 4.0.2 base package function “p.adjust”. 

## 3. Results

Of the 38,807 patients diagnosed with PTSD that were identified, only 5294 patients (4901 unique patients) were initiated on an antipsychotic after the diagnosis of PTSD, had no history of suicidal-related events prior to the enrollment, and no prior use of antipsychotics within one year, with at least one year of history in the electronic medical record. Eligible individuals were enrolled in this study and were assigned to a treatment group and followed until stopping current treatment, switching to another same class of drugs, death or loss to follow up. Quetiapine was the most commonly assigned antipsychotic, consisting of 29.2% of the study population. Lurasidone was the least commonly assigned, making up 4.26% of the population (Figure 1). Baseline characteristics are shown in Table 1.

Of the patients enrolled in this study, 5137 patients were right censored. The average follow up of the antipsychotics was 7.86 months, ranging from 4.38 to 9.76 months. Patients taking quetiapine (1429 patients) or aripiprazole (982 patients) experienced the most amount of SREs, as 38 events were recorded for both groups. Zero patients taking lurasidone experienced an SRE (Figure 2). 

Results of the emulations are shown in Table 2. Head-to-head comparisons are conducted between every pair of antipsychotics to compare their effectiveness in managing SREs. As shown in Table 2, if only adjusting for comorbidities (and age, gender, ED visits) lurasidone showed a significantly better effect compared to all other antipsychotics while ziprasidone and olanzapine were associated with higher SREs in patients.

To assess the effects of using concomitant medications on the SREs, we also adjusted for the most used medications. Those concomitant drugs are central nervous system agents and had been used at the baseline time in more than 5% of users of an antipsychotic drug of interest. To reduce number of variables in the models, those comorbidities with less than 5% patients in the paired antipsychotics were not included. After further adjusting those drugs, the hazard ratio of lurasidone against other antipsychotics are still significantly better with *p* values less than 0.0001 and adjusted p values less than 0.04 (except for perphenazine vs. lurasidone, which has no significant difference). We also noticed that aripiprazole vs. lurasidone has HR of 43.25 and haloperidol vs. lurasidone has HR of 52.20. These HRs have large confidence intervals (CI). This might indicate that the models become unstable when more covariates were added. Detailed information on all the concomitant medications can be found in Appendix C.

Figure 3 showed the standardized survival curves for lurasidone vs. olanzapine, risperidone vs. ziprasidone and olanzapine vs. risperidone. Please note, the curves were the results of adjusting covariates during the comparison of two paired drugs, and as such these curves do not necessarily reflect the real SREs listed above. The survival curves were generated according to the work of Danaei and colleagues [26]. We estimated survival curves for each arm by fitting a pooled logistic model and included product (“interaction”) terms to allow the hazard ratio to change over time.

## 4. Discussion and Conclusions

The primary objective of this observational study was to identify specific antipsychotics that may be associated with a decreased incidence of suicide-related events. In this study, we emulated a randomized controlled trial using the Neptune system at University of Pittsburgh with UPMC health system electronic medical records from January 2009 to October 2019. We utilized baseline characteristic data to assess suicide risk in patients diagnosed with PTSD. Patients were assigned one of the compared antipsychotics and the suicidal risk was statistically analyzed. 

Only a handful of studies have been conducted on antipsychotics for the treatment of PTSD, and even fewer on the medications’ effect on SREs [33]. As discussed, the results of the limited studies do show promising efficacy for the treatment of PTSD with SGAs [21,22]. Lurasidone, one of the less frequently prescribed antipsychotics, showed a statistically significant decrease in SREs compared to all the other antipsychotics: aripiprazole, haloperidol, olanzapine, quetiapine, risperidone, and ziprasidone (except perphenazine) (Table 1). In addition, olanzapine was associated with higher SREs than quetiapine and risperidone, and ziprasidone was associated with higher SREs than risperidone. The results of this study indicate that commonly used antipsychotics such as olanzapine may put individuals within the PTSD population at an increased risk of SREs. The findings of this study suggest that special considerations may need to be put in place whenever prescribing or recommending these medications to PTSD patients.

Antipsychotics have a variety of indications both FDA and non-FDA approved [34]. While the majority of medications in the class are indicated for schizophrenia and bipolar disorder, some other uses include Tourette’s disorder, nausea and vomiting, and chronic pain. Lurasidone is FDA approved for the treatment of schizophrenia as well as depression associated with bipolar disorder, or manic–depressive disorder, as adjunct therapy [35]. The findings of our study suggest that this antipsychotic may decrease the risk of SREs, and this information may be useful when recommending patient-specific pharmacotherapy to patients with PTSD. 

Ziprasidone, another less frequently prescribed antipsychotic, showed statistically significant higher SREs than risperidone. The atypical antipsychotic is FDA approved for the treatment of both bipolar disorder and schizophrenia. Hamner et al. conducted a randomized, 2-phase, placebo-controlled study of ziprasidone efficacy in refractory PTSD patients [36]. Patients refractory to SSRI treatment in phase 1 were assigned either ziprasidone or placebo. No statistically significant differences were discovered when ziprasidone was given to PTSD patients refractory to initial SSRI treatment, compared to placebo. As alternatives for PTSD treatment arise, pilot studies on antipsychotics such as this one are crucial. The efficacy and safety of ziprasidone in both the study by Hammer and company and our study are less than ideal for the target population. Prescribers considering alternate therapies for PTSD patients may want to take careful consideration of ziprasidone until further, larger, studies can be completed. 

Many studies and publications have covered antipsychotic drugs and the impact on SREs in schizophrenic patients, but the same relationship is scarcely trialed in the PTSD population [37]. Katz et al. conducted a study on suicidal risk in patients with both PTSD and bipolar disorder [38]. The multiple regression analysis identified that comorbid PTSD is a significant predictor and increases risk of suicidal ideation in bipolar patients. This study confirms the demand for more pharmacologic research for this patient population. With many antipsychotics indicated for the treatment of bipolar disorder, patients with comorbid PTSD may benefit from a specific antipsychotic that can lower the risk of SREs. If a physician is choosing an antipsychotic for a patient with PTSD, bipolar disorder, and a risk/history of SREs, recommending lurasidone may be a viable option. 

One of the other drawbacks of this study is the validity and usefulness of the lurasidone results. Though the medication displayed statistically significant fewer SREs than several more frequently prescribed antipsychotics, lurasidone had not been prescribed to PTSD patients since 2012 in the UPMC database systems. On the contrary, all the other antipsychotics included in this study showed an increase in enrollment since 2012. Though popular in the years of 2008 to 2012, lurasidone has had a sharp decline in use since that time period. One hypothesis to this observation points to modification in insurance company authorization. Lurasidone is an expensive medication with an estimated monthly cost of $1500. This high cost may have kept insurances wary of covering lurasidone as a first-line treatment before trial of several other less expensive antipsychotics. Insurance companies likely required the failure of two generically available antipsychotics before agreeing to cover lurasidone. Furthermore, most insurance companies would have likely required the diagnosis of schizophrenia or bipolar disorder. We further investigated the number of patients who switched their assignments and the number of ED visits at the first and the last month of follow ups. Our assumption is that if a drug is right censored because of its side effects or lack of therapeutic effects, the percentage of switching and/or number of ED visits would be higher than other drugs. Lurasidone users had comparable number of months of follow up, percentage of switching and average number of ED visits with users of other antipsychotics (Appendix D). We believe that the right-censoring of lurasidone is unlikely because of greater adverse effects or lack of efficacy.

Our study also has some limitations. Firstly, as this is not a randomized control study, we cannot rule out the possibility of uncontrolled biases. For example, socioeconomic status is an important factor for predicting SREs, but due to the limitation of our EMR dataset, such information is difficult to access. However, medication usage information was controlled which might mitigate the influence of some biases. In our recent publication, we predicted patients at increased risk for SREs with high accuracy through utilizing the demographic, comorbidities, and medication use information from EMR, without socioeconomic information [13]. Secondly, we defined SREs as ideation, attempts, and death by suicide. Because those three behaviors have very different constructs, it would be more suitable to use separate models [39]. However, in our study population of PTSD patients, which is a subpopulation of patients with high risk for SREs, the number of events is still low. To reach the statistical significance level, a larger number of patients is needed which is not feasible at this current stage. On the other hand, our study can be interpreted as the influence of antipsychotics on the risk shared among the ideations, attempts, and deaths by suicide. Thirdly, a great limitation of this work is the lack of any psychometric instrument concerning the severity of the symptoms of PTSD and lack of data on the motivation that led to the prescription of antipsychotics. Again, this is also related to the limitation of our EMR data and such information can be found in an unstructured data form such as medical notes which is a challenge to extract. 

Antipsychotics, in the context of treatment for PTSD, have typically been used as second-line agents as monotherapy or to augment the effects of first-line agents such as serotonin receptor inhibitors (SRIs). For patients who decline the use of antidepressants or psychotherapy for PTSD treatment, antipsychotics have emerged as a viable option. In our study, we controlled the use of antidepressants as two antidepressants paroxetine and sertraline are the first-line medications for PTSD. Our data showed that only a small portion of those patients were using those two medications at baseline. For example, the percentages of patients with coadministration of sertraline were in the range of 12.4% to 18.2% (Appendix C), while paroxetine usage was in the range of 1.0% to 5.2%. Limited studies have confirmed the effectiveness and safety of antipsychotics in the PTSD population, and there are currently no studies with lurasidone. With the current lack of knowledge and research of antipsychotics for PTSD patients, specifically lurasidone, we hope our study provides a critical gateway into the relationship between certain antipsychotics and suicide risk. 

## Figures and Tables

**Figure 1 jpm-11-00178-f001:**
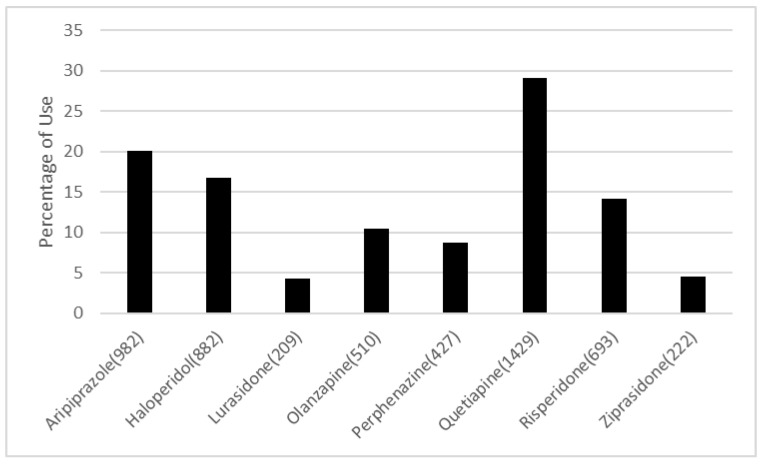
Percent of eligible PTSD patients receiving certain antipsychotics. As shown in the figure, no antipsychotics have the overall majority in treating eligible PTSD patients in our study (5294 eligible subjects from 4901 unique patients). Quetiapine, Aripiprazole and Haloperidol are the top three most used antipsychotics in PTSD patients.

**Figure 2 jpm-11-00178-f002:**
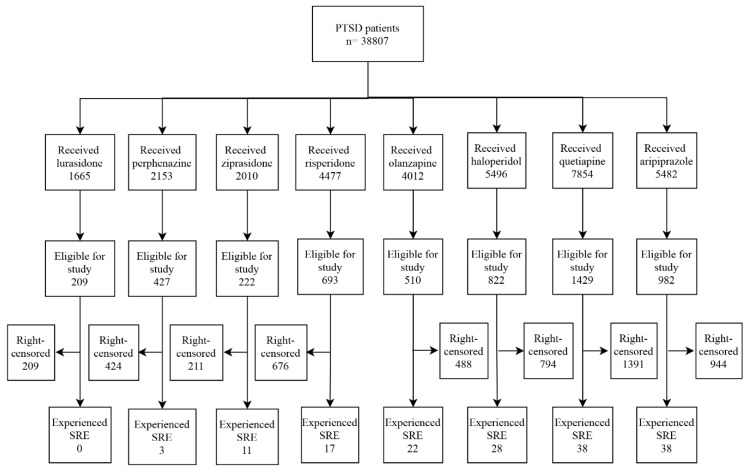
Enrollment process for emulation. Patients were right censored if they had no SREs at the end of the study period (i.e., administrative censoring), if a person failed to return for a study visit (i.e., lost to follow up), if they stopped using the antipsychotic of interest (no records of use within 3 months), or if they switched to another antipsychotic. SRE: suicide-related events.

**Figure 3 jpm-11-00178-f003:**
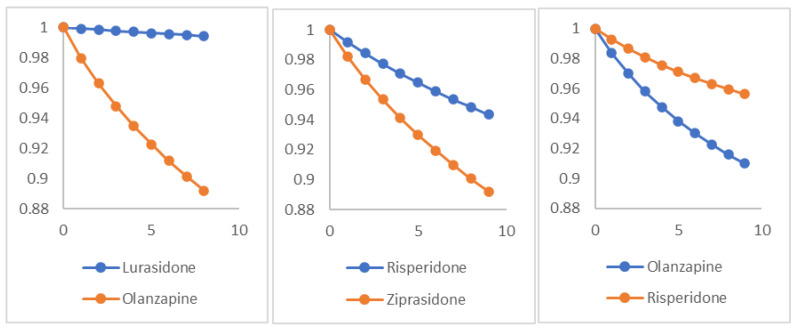
Examples of survival curves for three paired drugs on the outcome of SREs. *X* axis is months of follow up and *y* axis is the survival rate. These curves were generated by adjusting the baseline and post-baseline variables and may not reflect the real data. SREs: suicide-related events.

**Table 1 jpm-11-00178-t001:** Baseline characteristics.

	Level	Aripiprazole	Haloperidol	Lurasidone	Olanzapine	Perphenazine	Quetiapine	Risperidone	Ziprasidone	*p* Value
# of Patients		982	822	209	510	427	1429	693	222	-
Age (mean (SD))		38.13 (16.18)	46.48 (17.12)	37.71 (12.15)	42.64 (15.83)	42.71 (13.56)	40.31 (14.52)	38.39 (17.66)	37.24 (12.88)	<0.001
Male (%)		245 (24.9)	289 (35.2)	33 (15.8)	186 (36.5)	99 (23.2)	473 (33.1)	261 (37.7)	45 (20.3)	<0.001
# of ED visits in previous 3 months (mean (SD))		0.47 (1.26)	1.18 (2.11)	0.44 (0.90)	0.68 (1.31)	0.57 (1.06)	0.55 (1.08)	0.54 (1.20)	0.49 (0.88)	<0.001
Category 1 * (%)	0	876 (89.2)	549 (66.8)	189 (90.4)	421 (82.5)	390 (91.3)	1156 (80.9)	584 (84.3)	195 (87.8)	<0.001
	1	106 (10.8)	273 (33.2)	20 (9.6)	89 (17.5)	37 (8.7)	273 (19.1)	109 (15.7)	27 (12.2)	
Category 2 * (%)	0	957 (97.5)	788 (95.9)	201 (96.2)	482 (94.5)	423 (99.1)	1393 (97.5)	647 (93.4)	210 (94.6)	<0.001
	1	25 (2.5)	34 (4.1)	8 (3.8)	28 (5.5)	4 (0.9)	36 (2.5)	46 (6.6)	12 (5.4)	
Category 3 * (%)	0	408 (41.5)	187 (22.7)	104 (49.8)	176 (34.5)	137 (32.1)	540 (37.8)	284 (41.0)	93 (41.9)	<0.001
	1	574 (58.5)	635 (77.3)	105 (50.2)	334 (65.5)	290 (67.9)	889 (62.2)	409 (59.0)	129 (58.1)	
Category 4 * (%)	0	962 (98.0)	792 (96.4)	209 (100.0)	477 (93.5)	425 (99.5)	1400 (98.0)	653 (94.2)	216 (97.3)	<0.001
	1	20 (2.0)	30 (3.6)	0 (0.0)	33 (6.5)	2 (0.5)	29 (2.0)	40 (5.8)	6 (2.7)	
Category 5 * (%)	0	671 (68.3)	372 (45.3)	137 (65.6)	287 (56.3)	204 (47.8)	877 (61.4)	470 (67.8)	155 (69.8)	<0.001
	1	311 (31.7)	450 (54.7)	72 (34.4)	223 (43.7)	223 (52.2)	552 (38.6)	223 (32.2)	67 (30.2)	
Category 6 * (%)	0	951 (96.8)	781 (95.0)	202 (96.7)	489 (95.9)	419 (98.1)	1385 (96.9)	670 (96.7)	214 (96.4)	0.158
	1	31 (3.2)	41 (5.0)	7 (3.3)	21 (4.1)	8 (1.9)	44 (3.1)	23 (3.3)	8 (3.6)	
Category 7 * (%)	0	982 (100.0)	821 (99.9)	209 (100.0)	509 (99.8)	427 (100.0)	1428 (99.9)	692 (99.9)	222 (100.0)	0.881
	1	0 (0.0)	1 (0.1)	0 (0.0)	1 (0.2)	0 (0.0)	1 (0.1)	1 (0.1)	0 (0.0)	
Category 8 * (%)	0	980 (99.8)	821 (99.9)	208 (99.5)	509 (99.8)	425 (99.5)	1424 (99.7)	692 (99.9)	222 (100.0)	0.842
	1	2 (0.2)	1 (0.1)	1 (0.5)	1 (0.2)	2 (0.5)	5 (0.3)	1 (0.1)	0 (0.0)	
Category 9 * (%)	0	955 (97.3)	787 (95.7)	203 (97.1)	494 (96.9)	417 (97.7)	1383 (96.8)	679 (98.0)	216 (97.3)	0.343
	1	27 (2.7)	35 (4.3)	6 (2.9)	16 (3.1)	10 (2.3)	46 (3.2)	14 (2.0)	6 (2.7)	
Category 10 * (%)	0	964 (98.2)	753 (91.6)	207 (99.0)	475 (93.1)	422 (98.8)	1383 (96.8)	663 (95.7)	214 (96.4)	<0.001
	1	18 (1.8)	69 (8.4)	2 (1.0)	35 (6.9)	5 (1.2)	46 (3.2)	30 (4.3)	8 (3.6)	
Category 11 * (%)	0	905 (92.2)	764 (92.9)	197 (94.3)	466 (91.4)	414 (97.0)	1366 (95.6)	611 (88.2)	212 (95.5)	<0.001
	1	77 (7.8)	58 (7.1)	12 (5.7)	44 (8.6)	13 (3.0)	63 (4.4)	82 (11.8)	10 (4.5)	
Category 12 * (%)	0	979 (99.7)	816 (99.3)	209 (100.0)	506 (99.2)	426 (99.8)	1423 (99.6)	687 (99.1)	220 (99.1)	0.492
	1	3 (0.3)	6 (0.7)	0 (0.0)	4 (0.8)	1 (0.2)	6 (0.4)	6 (0.9)	2 (0.9)	

* 0 represents a patient not having a diagnosis that fits under the category; 1 represents a patient who does have a diagnosis that fits under the category (see Appendix B for more details). Numbers represent number of patients followed by the percentage of patients within each drug arm in parentheses unless otherwise specified. Chi-Squared Test [31] is used for categorical variables and Student *t* test [32] is used for continuous variables. SD: standard deviation; ED visits: emergency department visits; #: Number.

**Table 2 jpm-11-00178-t002:** Results of emulation of clinical trials on head-to-head comparison of antipsychotics for the outcomes of SREs with truncating weights.

Drug Pair	Hazard Ratio * 95% Confidence Interval (Adjusting Comorbidities)	*p* Value	Q Value **	Hazard Ratio 95% Confidence Interval (Adjusting Comorbidities + Concomitant Drugs)	*p* Value	Q Value **
Aripiprazole vs. Haloperidol	0.73 [0.437, 1.22]	0.233	0.2610	0.44 [0.25, 0.79]	0.0062	0.0154
Aripiprazole vs. Lurasidone	15.4 [8.65, 27.5]	<0.0001	<0.0004	43.25 [10.65, 175.56]	<0.0001	<0.0004
Aripiprazole vs. Olanzapine	0.70 [0.421, 1.15]	0.1599	0.2356	0.75 [0.41, 1.39]	0.3594	0.4193
Aripiprazole vs. Perphenazine	2.27 [0.847, 6.06]	0.1034	0.1756	1.09 [0.49, 2.44]	0.8319	0.8627
Aripiprazole vs. Quetiapine	1.41 [0.919, 2.17]	0.1159	0.1803	2.05 [1.28, 3.28]	0.0028	0.0087
Aripiprazole vs. Risperidone	1.37 [0.846, 2.22]	0.2002	0.2372	2.20 [1.21, 4.01]	0.0101	0.0202
Aripiprazole vs. Ziprasidone	0.68 [0.373, 1.23]	0.2033	0.2372	0.98 [0.52, 1.85]	0.9542	0.9542
Haloperidol vs. Lurasidone	23.2 [7.68, 70.2]	<0.0001	<0.0004	52.20 [14.97, 182.18]	<0.0001	<0.0004
Haloperidol vs. Olanzapine	1.05 [0.547, 2.00]	0.8938	0.9269	1.86 [1.02, 3.42]	0.0447	0.0834
Haloperidol vs. Perphenazine	4.25 [1.23, 14.7]	0.0219	0.0472	1.97 [0.80, 4.84]	0.1399	0.1996
Haloperidol vs. Quetiapine	1.73 [1.038, 2.87]	0.0354	0.0661	1.62 [0.86, 3.05]	0.1321	0.1996
Haloperidol vs. Risperidone	1.72 [0.89, 3.31]	0.1066	0.1756	3.20 [1.47, 6.97]	0.0033	0.0092
Haloperidol vs. Ziprasidone	0.590 [0.27, 1.30]	0.1897	0.2372	1.81 [0.82, 3.98]	0.1426	0.1996
Lurasidone vs. Olanzapine	0.057 [0.031, 0.10]	<0.0001	<0.0004	0.01 [0.002, 0.04]	<0.0001	<0.0004
Lurasidone vs. Perphenazine	0.13 [0.064, 0.28]	<0.0001	<0.0004	0.17 [0.01, 2.41]	0.1904	0.2423
Lurasidone vs. Quetiapine	0.12 [0.077, 0.18]	<0.0001	<0.0004	0.05 [0.02, 0.16]	<0.0001	<0.0004
Lurasidone vs. Risperidone	0.13 [0.073, 0.25]	<0.0001	<0.0004	0.24 [0.14, 0.40]	<0.0001	<0.0004
Lurasidone vs. Ziprasidone	0.065 [0.033, 0.13]	<0.0001	<0.0004	0.05 [0.02, 0.11]	<0.0001	<0.0004
Olanzapine vs. Perphenazine	2.945 [1.15, 7.58]	0.025	0.0500	0.66 [0.30, 1.43]	0.2901	0.3532
Olanzapine vs. Quetiapine	1.813 [1.104, 2.98]	0.0188	0.0439	1.95 [1.20, 3.15]	0.0066	0.0154
Olanzapine vs. Risperidone	2.153 [1.15, 4.02]	0.0161	0.0439	2.35 [1.41, 3.94]	0.0011	0.0039
Olanzapine vs. Ziprasidone	0.882 [0.43, 1.80]	0.7296	0.7857	1.61 [0.83, 3.11]	0.1572	0.2096
Perphenazine vs. Quetiapine	0.541 [0.22, 1.33]	0.1801	0.2372	1.22 [0.48, 3.11]	0.677	0.7582
Perphenazine vs. Risperidone	0.506 [0.18, 1.40]	0.188	0.2372	2.55 [1.27, 5.14]	0.0087	0.0187
Perphenazine vs. Ziprasidone	0.315 [0.12, 0.82]	0.0176	0.0439	1.93 [0.81, 4.59]	0.1358	0.1996
Quetiapine vs. Risperidone	1.024 [0.60, 1.75]	0.9286	0.9286	1.08 [0.64, 1.80]	0.7762	0.8359
Quetiapine vs. Ziprasidone	0.451 [0.24, 0.84]	0.0118	0.0367	0.59 [0.32, 1.08]	0.0866	0.1516
Risperidone vs. Ziprasidone	0.432 [0.23, 0.83]	0.0118	0.0367	0.30 [0.17, 0.52]	<0.0001	<0.0004

* Hazard ratio: the first drug vs. the second drug. Numbers represented in brackets indicate the 95% confidence intervals. ** Q values were generated by adjusting false discovery rate from *p* values.

## Data Availability

The data used in this study was from UPMC under a data use agreement. The authors are not allowed to distribute the data to any third party, but researchers may contact UPMC for data access.

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
