# Peer review of "An Emulation of Randomized Trials of Administrating Antipsychotics in PTSD Patients for Outcomes of Suicide-Related Events"

_jpm, 2021, doi:10.3390/jpm11030178_

Round 1

Reviewer 1 Report

The given manuscript by Delapaz et al. investigated the outcomes of patients suffering from post-traumatic stress disorder concerning suicide-related events. The authors used eletronic medical records of almost 39,000 patients
with post-traumatic stress disorder to investigate their suicide risk under several antipsychotic drugs. They found that lurasidone leads to a statistically significant decrease in suicide-related events, while olanzapine and ziprasidone lead to an increase when compared head-to-head to other drugs.
This study is well written and well discussed. Additionally, the results are of significant interest for the therapy of PTSD patients. I have one major remark concerning the presentation of the data and several minor remarks. After addressing these remarks, this manuscript is well suited for publication.

Major aspects
One of the study's results is that PTSD treatments using lurasidone resulted in 0 SREs. Given the fact that 0 SREs are a result of complete right-censoring, the reason of right censoring needs to be specified. 
The approach of right-censoring all patients not having SREs for their various reasons is statistically speaking a sound concept. Nevertheless, specifying the number of patients not having SREs among right-censored patients is of significant importance, since the current presentation of data may lead to the conclusion that lurasidone is an excellent drug to avoid SREs within the therapy of PTSD. However, the current presentation of data might also hide the fact that all of the patients stopped using it (e.g. due to adverse effects) or were unable to attend follow-ups (e.g. due to a significant decrease in their mental health). Ideally the fraction of each reason to be right-censored would be specified for all right-censored patients in all drug groups which would allow a reasonable interpretation of the data. However, it may be sufficient to specify the number of patients not having an SRE for the particular group of lurasidone patients.

Minor aspects
- Abstract: I am missing the usage of an emulation of randomized trials in the abstract. Since it is mentioned in the title, one would assume it is a vital characteristic of the study.
- Table 1: please define each abbreviation used within the table (SD, P, E, N), as well as the meaning of the numbers in parentheses.
- Figure 2: please define abbreviations used within the figure (SRB, SRE).
- Table 2: please define what the number in parentheses are, as well as abbrevtiations such as CI. I additionally noticed that numbers below 1 are written as 0.xxx whithout a less-than sign but as .xxx with it. I do not know if this is done intentionally but I'd recommend unifying it. 
- Page 10, line 280: "et al." is missing a ".".

Author Response

The given manuscript by Delapaz et al. investigated the outcomes of patients suffering from post-traumatic stress disorder concerning suicide-related events. The authors used eletronic medical records of almost 39,000 patients
with post-traumatic stress disorder to investigate their suicide risk under several antipsychotic drugs. They found that lurasidone leads to a statistically significant decrease in suicide-related events, while olanzapine and ziprasidone lead to an increase when compared head-to-head to other drugs.
This study is well written and well discussed. Additionally, the results are of significant interest for the therapy of PTSD patients. I have one major remark concerning the presentation of the data and several minor remarks. After addressing these remarks, this manuscript is well suited for publication.

Major aspects
One of the study's results is that PTSD treatments using lurasidone resulted in 0 SREs. Given the fact that 0 SREs are a result of complete right-censoring, the reason of right censoring needs to be specified. 
The approach of right-censoring all patients not having SREs for their various reasons is statistically speaking a sound concept. Nevertheless, specifying the number of patients not having SREs among right-censored patients is of significant importance, since the current presentation of data may lead to the conclusion that lurasidone is an excellent drug to avoid SREs within the therapy of PTSD. However, the current presentation of data might also hide the fact that all of the patients stopped using it (e.g. due to adverse effects) or were unable to attend follow-ups (e.g. due to a significant decrease in their mental health). Ideally the fraction of each reason to be right-censored would be specified for all right-censored patients in all drug groups which would allow a reasonable interpretation of the data. However, it may be sufficient to specify the number of patients not having an SRE for the particular group of lurasidone patients.

Response:

Thanks for your suggestions. It is very challenging to find the reasons of right-censoring from electronic medical records. However, we tried to find some indirect evidence to exclude the possibility of adverse effects or therapeutic failure by looking into the percentage of patients switching assigned drugs and the number of emergency department visits at the first and the last months to see if there are any clues. We found that these numbers are comparable among lurasidone users and other antipsychotics. The following changes have been made in the discussion section (lines 304-309)

We further investigated the number of patients who switched their assignments and the number of ED visits at the first and the last month of follow-ups. Our assumption is that if a drug is right-censored because of its side effects or lack of therapeutic effects, the percentage of switching and/or number of ED visits would be higher than other drugs. Lurasidone users had comparable number of months of follow-up, percentage of switching and average number of ED visits with users of other antipsychotics (Appendix D). We believe that the right-censoring of lurasidone is unlikely because of greater adverse effects or lack of efficacy.”

Minor aspects
- Abstract: I am missing the usage of an emulation of randomized trials in the abstract. Since it is mentioned in the title, one would assume it is a vital characteristic of the study.
- Table 1: please define each abbreviation used within the table (SD, P, E, N), as well as the meaning of the numbers in parentheses.
- Figure 2: please define abbreviations used within the figure (SRB, SRE).
- Table 2: please define what the number in parentheses are, as well as abbrevtiations such as CI. I additionally noticed that numbers below 1 are written as 0.xxx whithout a less-than sign but as .xxx with it. I do not know if this is done intentionally but I'd recommend unifying it. 
- Page 10, line 280: "et al." is missing a ".".

Response:

The corrections have been made according to your suggestions. Thanks.

Reviewer 2 Report

This study reflects a good example of using observational data to make critical conclusions. It is a well designed and well executed study with appropriate statistical analysis showing important association of antipsychotics and SRE's.

The results and discussion are clearly presented.

Author Response

Thanks for your nice comments.